# Classification of User Behavior Patterns for Indoor Navigation Problem

**DOI:** 10.3390/s25154673

**Published:** 2025-07-29

**Authors:** Aleksandra Borsuk, Andrzej Chybicki, Michał Zieliński

**Affiliations:** Faculty of Electronics, Telecommunications and Informatics, Gdańsk University of Technology, ul Narutowicza 11/12, 80-233 Gdańsk, Poland; s184620@student.pg.edu.pl (A.B.); s184372@student.pg.edu.pl (M.Z.)

**Keywords:** indoor navigation, activity classification, LSTM, sensor data, linear acceleration, sensor fusion data, mobile application

## Abstract

Indoor navigation poses persistent challenges due to the limitations of traditional positioning systems within buildings. In this study, we propose a novel approach to address this issue—not by continuously tracking the user’s location, but by estimating their position based on how closely their observed behavior matches the expected progression along a predefined route. This concept, while not universally applicable, is well-suited for specific indoor navigation scenarios, such as guiding couriers or delivery personnel through complex residential buildings. We explore this idea in detail in our paper. To implement this behavior-based localization, we introduce an LSTM-based method for classifying user behavior patterns, including standing, walking, and using stairs or elevators, by analyzing velocity sequences derived from smartphone sensors’ data. The developed model achieved 75% accuracy for individual activity type classification within one-second time windows, and 98.6% for full-sequence classification through majority voting. These results confirm the viability of real-time activity recognition as the foundation for a navigation system that aligns live user behavior with pre-recorded patterns, offering a cost-effective alternative to infrastructure-heavy indoor positioning systems.

## 1. Introduction

Recognizing human behavior through smartphone sensor data is a growing area of interest in mobile computing, with important applications in navigation, healthcare, smart environments, and logistics. This study presents a novel approach to behavior recognition using only linear acceleration sensor data from smartphones, with the goal of supporting future indoor navigation systems. Unlike traditional indoor navigation methods, which primarily rely on dead reckoning or positioning [1,2], the proposed solution focuses on analyzing and tracking user behavior—such as standing, walking, climbing stairs up/down, or taking elevators up/down—based solely on mobile sensor data [3]. Through a series of experiments, we demonstrate that linear acceleration data can differentiate between these six activity types, enabling a new way to track progress through indoor environments without requiring maps or GPS.

The primary motivation for this research stems from the need to improve the efficiency of delivery services, particularly in apartment buildings and residential complexes, where finding a specific apartment is often time-consuming. While the challenges of “door-to-building” navigation—using satellite techniques or popular GPS solutions—have been relatively well-documented in the literature and implemented in industrial practice [4,5], effective navigation within buildings (from the entrance to the target location) still remains a challenge [6]. In practice, the problems of locating a specific apartment—including identifying the correct floor, choosing the optimal path, and navigating through complex hallway layouts—largely rely on the courier’s prior experience or manually provided instructions, such as “second floor, stairs on the left, last apartment in the hallway”. Both methods require additional time to familiarize oneself with the route or formulate instructions, leading to delays and increased operational costs.

A key difficulty in developing a universal indoor navigation system lies in the lack of easily accessible positioning solutions (e.g., due to a limited GPS signal range [2,7] or the financial costs of installing infrastructure [8] for indoor localization) as well as the impracticality of maintaining detailed 2D or 3D maps of buildings [9]. To address these limitations, we propose a concept for indoor navigation that utilizes behavioral pattern recognition based on smartphone linear acceleration sensor data [10]. The key assumption is the analysis of paths taken by couriers or delivery personnel who have previously traversed a specific route within the building. For example, if courier X1 reaches apartment X in building Y, the system will record and process the history of their movement and characteristic behavior patterns. When courier X2 later heads to the same apartment, the system guides them by aligning their real-time movement data with the stored trajectory, providing precise guidance.

Most prior activity recognition studies rely on multi-sensor fusion (accelerometer, gyroscope, magnetometer, barometer) [11] or on additional wearable devices (e.g., sensors placed on the ankle) [12] to improve classification accuracy. While these approaches achieve strong results, they often demand more power, calibration, or hardware complexity, limiting their practicality for low-cost, large-scale deployment. In contrast, our method focuses on identifying a single sensor that is universally available on smartphones and requires no calibration. We show that, even with this single modality, it is possible to recognize indoor movement behaviors relevant to navigation tasks with high accuracy. This minimalist design reduces computational overhead and sensor dependency while maintaining sufficient classification performance for real-world applications in logistics and delivery.

## 2. Techniques Applied in Indoor Navigation

Indoor navigation refers to various techniques and algorithms designed to track and position users within enclosed spaces such as shopping malls, office buildings, hospitals, and apartment complexes [13]. Due to the limited coverage and accuracy of satellite-based systems (e.g., GPS) in indoor environments [14], several alternative solutions have been developed. Some of these rely on radio-based infrastructure (Wi-Fi [15,16], Bluetooth/BLE [17,18]), while others utilize natural environmental features (e.g., magnetic field measurements [19,20,21]) or motion sensors embedded in smartphones (accelerometers, gyroscopes, magnetometers) [22]. Typical design challenges include ensuring accurate localization, managing supporting infrastructure (e.g., access points), and adapting the system to diverse building layouts. The literature emphasizes that there is no universal, fully effective method of indoor navigation—each system must be tailored to specific needs and environmental conditions [23]. In this section, we briefly review several prominent approaches to indoor navigation.

### 2.1. Inertial Navigation

One of the simplest methods for indoor localization relies on a smartphone’s built-in accelerometer to count steps (so-called step detection) and determine traveled distance [24]. This technique, often referred to as step-counting navigation, involves identifying characteristic acceleration patterns associated with walking. By counting detected steps and using an estimated step length—typically based on the average human stride—the system approximates the distance covered. The main advantage of this method is that it requires no additional infrastructure within the building, relying solely on the smartphone’s accelerometer sensor. However, its accuracy is affected by walking irregularities, variations in step length, and differences in phone placement (e.g., in hand, pocket, or bag) [25,26]. Therefore, the literature suggests applying signal filtering techniques and advanced machine learning methods for step detection and classification [27].

### 2.2. Beacon/BLE Navigation

In recent years, localization systems based on Bluetooth Low Energy (BLE) technology have gained popularity. These systems involve placing small transmitters, known as beacons, throughout a building, which periodically broadcast signals with a unique identifier [28]. A smartphone application, upon receiving these signals and measuring their strength (Received Signal Strength Indicator, RSSI), can estimate the user’s approximate distance from a particular transmitter. By using multiple beacons placed in known locations and applying triangulation or trilateration, it is possible to determine the user’s current position. While BLE-based systems generally offer higher accuracy than step detection, they require the installation and maintenance of beacon infrastructure, leading to financial and organizational costs [29]. Additionally, their performance may be affected by environmental factors such as obstacles, metal objects, crowd density, or radio interference from other devices.

### 2.3. Wi-Fi Navigation

In addition to deploying dedicated beacons, it is also possible to utilize the existing Wi-Fi infrastructure for indoor positioning. Beyond the traditional measurement of the Received Signal Strength Indicator (RSSI), which is often affected by signal fluctuations, modern smartphones can estimate the time it takes for a signal to travel between the device and access points using Wi-Fi Round-Trip Time (RTT), reducing the impact of environmental interference and providing greater precision and stability across different building layouts and smartphone models [15]. Positioning accuracy can further be enhanced by combining RSSI-based distance estimates with optimization techniques such as Particle Swarm Optimization (PSO), which prioritizes measurements from closer access points, helping mitigate errors caused by signal interference and reflections [16]. Nevertheless, Wi-Fi-based navigation still faces several challenges, including the need for device and infrastructure compatibility, periodic recalibrations, sensitivity to environmental changes, and, in the case of optimization-based methods, higher computational complexity.

### 2.4. Geomagnetic Field Fingerprinting Navigation

Another approach to indoor localization leverages the natural variations in the Earth’s magnetic field, which can be disrupted by steel structures and electrical devices within buildings [30]. This technique, commonly known as magnetic field fingerprinting, involves measuring the magnetic field’s strength and direction at various points inside the building (the calibration phase) and then comparing real-time readings from the smartphone’s magnetometer to these pre-recorded patterns (known as fingerprints). In practice, this allows for estimating the user’s location without requiring additional infrastructure. However, the method is sensitive to environmental changes, such as rearrangements or the introduction of new electrical devices and can be prone to calibration errors. Despite these challenges, studies indicate that magnetic field-based localization can achieve an accuracy of less than 1 m in typical office environments [31].

### 2.5. Dead Reckoning Navigation

Beyond simple step counting, a broader category of indoor navigation methods known as dead reckoning (DR) has been developed. This approach involves continuously updating the user’s position based on data from multiple inertial sensors (accelerometer, gyroscope, and sometimes a barometer), combined with signals from a magnetometer to determine orientation [32]. These techniques are often supported by filtering methods (e.g., Kalman filter, particle filter) to minimize accumulating measurement errors. Compared to simple step counting, DR is more flexible and can account for changes in the phone orientation, different walking styles, and turns. However, a major challenge that remains is error accumulation over longer distances—even small deviations at each step can add up, leading to significant inaccuracies over several hundred meters. Therefore, the literature often suggests combining DR with other localization techniques (e.g., BLE beacon measurements or magnetic field patterns) in so-called hybrid systems [33].

### 2.6. Pattern Based Approaches

Although less commonly used, several additional human activity recognition (HAR) techniques can also be found in the literature. While most HAR research focuses on applications in healthcare monitoring and assessment, to some extent, similarities can be observed between these techniques and the methods explored in our study. The field encompasses various families of methods leveraging smartphone-integrated sensors.

Firstly, traditional approaches rely on manually engineered features extracted from segmented raw sensor data, including time-domain metrics (such as mean and variance) and frequency-domain representations (e.g., FFT—Fast Fourier Transform). These features serve as inputs to classical classifiers like Support Vector Machines, Random Forests, or k-Nearest Neighbors, offering interpretability and low computational cost, though their effectiveness largely depends on the quality of the engineered features [34].

Secondly, deep-learning methods such as Convolutional Neural Networks (CNNs) and Long Short-Term Memory (LSTM) networks have gained popularity due to their ability to learn complex representations directly from raw sensor data. Models like convolutional LSTM architectures have achieved high classification accuracy (exceeding 90%) on standard datasets (e.g., WISDM, UCI-HAR) and have proven robust across both static and dynamic activity recognition tasks [35].

Thirdly, hybrid approaches combine both paradigms: for example, using dimensionality reduction techniques (e.g., Principal Component Analysis, PCA, or Kernal Principal Component Analysis, KPCA) or hand-crafted features as input into deep models to reduce computational cost while maintaining high accuracy [36].

Finally, context-sensitive multi-sensor fusion enriches inertial data with auxiliary information—such as barometers for altitude change, GPS for location, or light/proximity sensors to detect phone placement (e.g., pocket versus hand)—to improve recognition in real-world conditions [37]. Advanced trends also include self-supervised learning, federated learning, and edge-based models to reduce privacy concerns and power consumption, as noted in recent systematic reviews [38].

## 3. Proposed Solution

### 3.1. Formal Description

Formally, the task of matching real-time mobile phone signals to a predefined route pattern can be modeled as a sequence segmentation or sequence matching problem. Let the predefined route be represented by a sequence of *n* expected behavioral events:T = ⟨(e_1_, p_1_), (e_2_, p_2_), … , (eₙ, pₙ)⟩,(1)
where each event e_i_, i ∈ {1, …, n} represents a specific type of user behavior (e.g., walking, turning, waiting, elevator use, stair climbing), and p_i_ denotes a set of parameters describing its expected features, such as mean duration, step frequency, or number of floors or stairs traversed.

As the user follows the route, the mobile system continuously collects sensor data, producing a sequence of observations:X = ⟨x_1_, x_2_, … , xₘ⟩,(2)
with x_t_, t ∈ {1, 2, …, m} being an m-element vector of sensor readings (e.g., from the accelerometer or gyroscope) captured at time t.

To assess how well an observation corresponds to a given event, a scoring function is defined:δ(e_i_, x_t_; p_i_),(3)
which quantifies how closely the observation matches the expected behavior.

In the basic approach, each observation is assigned to the event with the highest score:ê_t_ = argmax (δ(e_i_, x_t_; p_i_)) where e_i_ ∈ {e_1_, …, eₙ},(4)
resulting in a predicted event sequence:Ê = ⟨ê_1_, ê_2_, …, ê_m_⟩(5)
This sequence can then be compared with the reference pattern *T* to determine which part of the route the user is currently traversing.

To maintain consistency with the predefined event order, a monotonicity constraint is enforced:ê_t+1_ ≥ ê_t_(6)
ensuring that the predicted sequence respects the natural forward progression of the user along the route and preventing regressions to earlier stages.

This framework supports various implementation strategies, including Dynamic Time Warping (DTW), Hidden Markov Models (HMMs) or LSTM-based classifiers. Each method can approximate or optimize the segmentation process in real time. By computing the predicted event sequence Ê online, incoming sensor data is segmented into behaviorally meaningful units, enabling robust applications in activity recognition, indoor navigation, and context-aware services.

### 3.2. The Role of Pattern Matching in Behavior

The proposed solution focuses on analyzing smartphone sensor signals to identify specific user activities. By utilizing data from the linear accelerometer in an Android phone, the system recognizes behavioral patterns such as walking, climbing stairs, descending stairs, standing, and elevator usage (ascending and descending). A key step in this process was identifying characteristic features for each of these activities. To achieve this, we collected, compared, and analyzed data from multiple sensors to determine correlations between them. After evaluation, we concluded that data from the linear acceleration sensor best meets the system’s requirements.

Pattern matching is performed by comparing current sensor data with previously recorded behavioral patterns. For this purpose, we deployed advanced machine learning algorithms, particularly the Long Short-Term Memory (LSTM) network, which is highly effective at analyzing sequential data [39,40].

The primary advantage of this approach is its universality—the system does not require additional infrastructure, such as markers or building maps, as it solely relies on data from mobile devices.

### 3.3. Developed Testbeds

As part of our research on predicting user behavior based on sensor data, we designed and implemented a dedicated test environment. Its purpose was to collect data from mobile devices for particular types of activity in order to perform the analysis of user behavior in the context of indoor navigation and to assess the feasibility of predicting physical activities based on sensor readings.

To gather data, we developed a mobile application in Android Studio, called DataCollector [41], that interacts with sensors using the SensorManager class [42]. To ensure a high sampling rate, we declared the HIGH_SAMPLING_RATE_SENSORS permission, enabling access to sensor data at frequencies exceeding 200 Hz (i.e., measurements taken at intervals shorter than 5 milliseconds).

Before data collection, the user selects a specific activity to be recorderd during that test (data labeling). The application then records data from multiple sensors, including the accelerometer, gyroscope, linear acceleration, magnetic field, rotation vector, and gravity sensor, and annotates the recorderd data by the user activity type. The collected data, together with the registered exact timestamp, is transmitted to a server and stored in a database. To prevent data loss due to connectivity issues—such as those occurring in an elevator—a caching mechanism is implemented, temporarily storing data and transmitting it once a connection is restored. The genral architectiure of this solutions is presented in Figure 1.

Figure 2 illustrates the application’s workflow. First, a test conductor enters a brief description for the test session, e.g., “House”, and then selects an activity. Once the test is completed, the conductor can either save the collected data or discard it if the test was not executed correctly. This approach helps minimize the chances of storing inaccurate or irrelevant data in the database.

### 3.4. Input Data Description and Utilized Technologies

The Android operating system allows device manufacturers to integrate various sensors that measure specific physical parameters related to smartphones. According to the Android documentation [42,43], an Android device can include up to ten motion sensors, six position sensors, and five environmental sensors. Although the documentation does not specify strict quality requirements for each sensor, device manufacturers must comply with the Android Compatibility Definition Document (CDD), which outlines minimum hardware and software standards, including measurement accuracy, update frequency, measurement range, and minimum latency.

Access to sensor data is provided through the Sensor API, a component of the Android SDK. The Sensor API allows developers to interact with sensors via the SensorManager class, responsible for managing access to the device’s sensors. Developers can retrieve a list of available sensors and register listeners for sensor data by implementing the SensorEventListener interface [44], which includes methods such as onSensorChanged() and onAccuracyChanged(), triggered when sensor data or its accuracy changes.

It is important to note that Android devices may be equipped with varying sets of sensors. Their number and accuracy depend on the manufacturer, model, and hardware class. Budget models often feature a limited set of sensors, while high-end flagship devices may include more precise and sophisticated sensors, such as barometers, geomagnetic sensors, or advanced accelerometers. Therefore, applications utilizing the Sensor API should dynamically check for the availability of specific sensors on a given device to ensure optimal compatibility.

In this study, we conducted experiments using sensor data commonly referenced in the inertial navigation literature, aiming to determine whether specific sensors could effectively distinguish between different activities. The following sensors were utilized:
Linear acceleration sensor—measures acceleration along the x, y, and z axes (m/s^2^), excluding the influence of gravity. This ensures that only motion-related forces are considered, improving activity recognition. It is computed aslinear acceleration = accelerometer reading − gravity component(7)Accelerometer sensor—measures total acceleration, including gravity, along the x, y, and z axes (m/s^2^). It is widely used for detecting movement and changes in orientation.Gravity sensor—measures the force of gravity (m/s^2^) along the x, y, and z axes, providing information about device tilt and orientation relative to Earth. The gravity sensor uses the accelerometer as a baseline but filters out dynamic forces, such as phone acceleration (e.g., hand movement) or vibrations.Gyroscope sensor—measures angular velocity (rad/s) around the x, y, and z axes. It is essential for detecting rotational movements and orientation changes.Magnetic field sensor—detects geomagnetic field strength (μT) along the x, y, and z axes. It is primarily used for compass-based navigation and orientation determination.Rotation vector sensor—combines accelerometer, gyroscope, and magnetometer data to estimate device orientation. It is particularly useful in applications requiring smooth and continuous motion tracking. It outputs rotation vector components (x, y, and z) and a scalar component based on quaternion representation.

For efficient data management, each sensor type was assigned a shortened name and a dedicated database table. In total, six tables were created: accelerometer, gyroscope, magnetic_field, gravity, linear_acceleration, and rotation_vector.

Additionally, each recorded activity was labeled with a unique tag to support model training:Ten steps—walking 10 steps in a straight line;Elevatordown—descending one floor in an elevator;Elevatorup—ascending one floor in an elevator;Stairsdown—descending five stairs;Stairsup—ascending five stairs;Standing—standing still.

These labels serve as class identifiers for training the LSTM model used for activity classification.

## 4. Methodology and Experiments

### 4.1. General Overview of Methodology

The initial stage of the research focused on identifying distinctive features for each activity. After collecting the data, we analyzed it by comparing readings from different activities for each sensor type, aiming to identify a sensor capable of clearly distinguishing between them. Given that sensor data was collected irregularly and varied by device, we applied linear interpolation to standardize it to fixed 2 ms intervals, a resolution chosen to balance the high-frequency sampling, often as fast as 1 millisecond, with the need to maintain consistent data for analysis. Our findings indicated that raw data alone might be insufficient for unequivocal activity classification. Therefore, to support statistical analysis using machine learning techniques, we unified the representation of sensor signals by adjusting scales without explicitly considering units. This approach emphasized the relative dynamics in the data, allowing us to more effectively detect similarities and differences in signal patterns across various sensor types.

The linear acceleration sensor provided clear signals related to movement, but the patterns observed for different activities were often similar, making it challenging to distinguish between, e.g., walking, ascending, and descending stairs (Figure 3).

The accelerometer data further confirmed this challenge. Although step signals were clearly visible, the overall signal shape and amplitude across walking and stair-related activities were very similar, complicating effective classification (Figure 4).

The gravity sensor data showed no clear relationship between the activity classes, making it unsuitable for classification (Figure 5).

The gyroscope exhibited excessive fluctuations in recorded values, significantly complicating the analysis and interpretation of the results (Figure 6).

The magnetometer, although in some cases capable of clearly differentiating between activities (Figure 7), proved to be highly sensitive to external factors. During our tests, we noticed disruptions in readings near elevators or large metal structures. These environmental interferences significantly reduced the reliability of the magnetometer, leading us to conclude that it is not suitable for consistent activity classification in indoor environments.

Lastly, the rotation vector data varied depending on the test, making it difficult to reliably assign them to a specific activity (Figure 8).

Based on these observations, we concluded that additional data processing methods were necessary to extract more distinctive features for each analyzed activity.

Since the most consistent and distinct data was recorded by the accelerometer and linear acceleration sensor, we considered it valuable for further analysis. The next step involved defining methods for processing this data to extract meaningful descriptors that would better align with the research requirements. To achieve this, we calculated the velocity for each of the x, y, and z axes using the equation:v = v_0_ + at,(8)
where v_0_ is the velocity computed in the previous measurement, a is the current acceleration measured by the sensors, and t is the time interval between measurements. As a result, we obtained data that enabled a more accurate interpretation of the activities.

The analysis revealed that the velocity accumulation calculated from accelerometer data showed a steady increase over time, which did not provide significant value for classification (Figure 9).

However, for the linear acceleration sensor data, we observed clearer patterns for each activity in the z-axis. The plots showed that during standing, the values oscillated around zero, while in the elevator, the z-axis acceleration component changed in the opposite direction depending on the movement (up or down). Stair climbing was characterized by consistent changes in the z-axis values, whereas descending stairs and walking on a flat surface exhibited similar, slightly increasing trends. In the case of descending stairs, the values rose more rapidly, allowing them to be differentiated from walking (Figure 10).

To ensure that the data was representative, we collected it using two phones independently (Samsung Galaxy M23 5G and Samsung Galaxy A45 5G, Samsung Electronics Co., Ltd., Suwon, Republic of Korea), carried by two participants in various environments, such as the Gdańsk University of Technology buildings and staircases in residential blocks in Gdańsk. Although the graphs showed slight differences depending on the phone or type of elevator, the overall shape of the plots remained similar. Figure 11 shows example linear accelerometer z-axis data, illustrating variations caused by different elevator types—one test was conducted in a university elevator, the other in a residential block elevator.

The clearly distinguishable behavioral patterns between activity categories, combined with consistent plot shapes within each category (for instance, Figure 10a vs. Figure 10b or Figure 10d vs. Figure 10e) confirmed that processed data from the z-axis of the linear acceleration sensor could serve as a solid foundation for further activity analysis and classification. Moreover, these patterns were repeatedly observable in more tests performed during the study.

### 4.2. Conducted Experiments

To evaluate the effectiveness of the z-axis velocity derived from the linear accelerometer sensor for activity classification, we conducted a series of experiments using six predefined activities: ten steps, elevator down, elevator up, stairs down, stairs up, and standing. For each activity, we recorded 80 trials resulting in a total dataset of 480 sequences. The dataset was then split into three subsets:For training, 70% (56 samples per activity);For validation, 15% (12 samples per activity);For testing, 15% (12 samples per activity).

The sensor data was collected with irregular timestamps and varied between devices. To standardize the dataset, we applied linear interpolation to resample the data at uniform 2 ms intervals. Additionally, we derived the velocity feature based on acceleration values over time, as it was a key feature for activity classification, using the modified formula described in Equation (8):v_i_ = v_i−1_ + a_i_ × Δt,(9)
where v_i_ is the velocity computed in the previous measurement, a_i_ is the current acceleration measured by the sensors, Δt is the time interval between consecutive measurements, and i denotes the measurement index. After preprocessing, we segmented the data into overlapping time windows of 500 samples (corresponding to 1 s), with a step size of 100 samples (200 ms), ensuring sufficient context for the LSTM model, and scaled it using the Standard Scaler to standardize the data distribution, which is a common preprocessing step for LSTM networks.

### 4.3. Utilized Algorithms

For activity classification, we implemented a Long Short-Term Memory (LSTM) neural network using TensorFlow and Keras [45]. LSTMs are well-suited for sequential data analysis, as they can capture long-term dependencies in time series data [40]. Our model architecture consisted of

An input layer with a shape corresponding to the window length (500 samples) and one feature (velocity), representing sequential sensor measurements over time.Two LSTM layers, the first with 128 units and the second with 64 units. The first layer outputs a sequence of hidden states (one per sample), which serve as input to the second layer. The second layer produces a single hidden state vector that summarizes the temporal dependencies in the input sequence.A dense layer with 4 neurons and ReLU activation designed to learn non-linear feature representations from the LSTM outputs.An output layer with a softmax activation function that converts the 64-dimensional dense layer output into a probability distribution over six activity classes, enabling sequence classification.Dropout layers (10%) after each LSTM layer and the dense layer to prevent overfitting and improve model generalization.

The model was compiled using the Adam optimizer and a sparse categorical cross-entropy loss function with a batch size of 32. To prevent overfitting, we implemented early stopping with a patience of 10 epochs, meaning training was halted if the validation loss did not improve for 10 consecutive epochs.

After training the model, we visualized the training and validation loss over epochs to identify potential overfitting. The best model was selected based on the epoch with the lowest validation loss. Figure 12 illustrates the changes in the training and validation loss, highlighting the point at which the validation loss was minimized. This figure shows that both training and validation loss decreased steadily without significant divergence, suggesting that the model did not overfit and generalized well to unseen data.

## 5. Results

We evaluated our model using two classification strategies: window-based and full-sequence. In the window-based approach, we classified each 1 s segment of data independently. In contrast, the full-sequence method aggregated overlapping predictions using majority voting, allowing us to produce more stable results.

While the window-based classification yielded 75% accuracy, the full-sequence approach significantly improved performance, reaching 98.6% by effectively reducing short-term noise and fluctuations.

To better understand the model’s behavior, we generated confusion matrices—Figure 13 shows the results for window-based classification, while Figure 14 presents those for full-sequence prediction.

### 5.1. Results for Particular Test Cases

Upon analyzing specific test cases, we observed that most misclassifications in the window-based approach occurred between elevator-related activities and standing. Of all elevator test samples, 22% were classified as standing, 12% of elevator down instances were classified as elevator up, and 25% of elevator up instances were classified as elevator down.

One possible explanation for the elevator misclassifications is the variability in elevator movement profiles. While all elevators generally follow the same pattern—an initial acceleration phase followed by deceleration—the exact characteristics (e.g., acceleration rate, stopping duration) may vary slightly depending on the elevator system. These variations may make it more challenging for the model to distinguish between elevator up and elevator down based on short, isolated windows.

Another contributing factor is the similarity between elevator up and elevator down movement profiles. For example, elevator up sequences typically exhibit an initial upward acceleration followed by deceleration, while elevator down shows the opposite. Some windows from elevator up may resemble the later windows of an elevator down sequence, leading to confusion.

Moreover, during elevator rides, noticeable velocity changes occur mainly when the elevator is accelerating or slowing down. The rest of the time, because the user remains inside the elevator, the data can resemble a standing pattern. This likely explains some of the misclassifications as standing.

Full-sequence classification significantly reduced these errors, as expected, resulting in near-perfect activity classification. However, confusion between elevator down and standing persisted, with one misclassification recorded. This suggests that while majority voting improves accuracy by incorporating a broader temporal context, distinguishing between these two activities remains challenging in certain cases.

Figure 15 illustrates the z-axis linear acceleration velocity for a misclassified elevator down sequence. Although the plot resembles the expected pattern for its class, the signal during elevator movement is short and weak, and the signal remains mostly flat. This lack of variation may have contributed to its misclassification as standing.

While our approach does not guarantee perfect classification, the results indicate that using an LSTM network to analyze processed data from a linear accelerometer sensor is a promising method for activity classification. The substantial improvement in accuracy achieved through full-sequence classification highlights the effectiveness of leveraging temporal dependencies in sequential data.

### 5.2. Real-Case Verification of Prosposed Approach

To validate the effectiveness of the proposed activity classification method, we conducted a practical experiment simulating a typical indoor navigation scenario. The goal was to evaluate the system’s ability to recognize user behavior patterns in a real-world setting using only background sensor data.

The test was carried out by Authors at the Faculty of Electronics, Telecommunications, and Informatics of Gdańsk University of Technology using a Samsung Galaxy M23 smartphone (Samsung Electronics Co., Ltd., Suwon, Republic of Korea). The predefined route included a sequence of diverse indoor movements:**Walking:** Walked nine steps along a hallway leading to the staircase.**Stair Climbing:** Climbed one floor via two stair segments.**Walking:** Walked 11 steps toward the elevator.**Standing:** Stood still while waiting for the elevator to arrive.**Entering the elevator:** Took three steps to enter the elevator.**Elevator ascent:** Rode the elevator two floors up.**Exiting elevator:** Walked 12 steps forward after leaving the elevator.

Each activity phase was individually timed, yielding the following durations: 7.26 s (walking), 6.77 s (stair climbing), 7.12 s (walking), 8.88 s (standing), 2.95 s (entering elevator), 14.40 s (elevator up), and 7.86 s (exiting elevator).

After data collection, we applied our trained LSTM model to predict activity labels for each non-overlapping one-second segment of the sequence. Figure 16 illustrates the classification results across the timeline of the route. In this section, we deliberately focused on a single example route to demonstrate a use case resembling a practical scenario—such as in courier deliveries—where a short navigation sequence lasting approximately 60 s may consist of several different types of user activities. Each colored segment represents the predicted activity for a 1 s window, while the stage markers at the bottom indicate the actual activities along the route. For visualization clarity, velocity values in the plot are shown as incremental, although the actual model input for each prediction window was normalized to start from zero.

In Stage 1, the model accurately recognized the walking activity, with a few misclassifications at the beginning and end. The first one-second window was labeled as stairs down, while the last two windows were classified as stairs down and stairs up, respectively. These errors may result from similarities in the signal patterns, which can lead to occasional minor misclassifications. Stages 2 and 3, both representing stairs up, were classified almost perfectly, with only one error occurring in the middle of Stage 2. This suggests that stair ascent is distinctly recognized by the model and well-differentiated from other activities. In Stage 4, which, like Stage 1, represents walking, the model behaved similarly—performing well overall—with only the final seconds misclassified: once as stairs down and once as stairs up. Stage 5, which involved standing still, was initially misclassified as elevator movement. This confusion is understandable, as standing in an elevator and standing still can produce very similar sensor readings. The performance dropped in Stage 6, where the user took only three steps. The beginning of this stage was classified as elevator down, possibly because the one-second window included data from the preceding stage (Stage 5). The end was incorrectly labeled as stairs down. The model seems to frequently confuse the end of walking with stair-related movements. Stage 7, involving a two-floor elevator ascent, showed the most classification errors. The model struggled to differentiate between standing inside the elevator and standing outside of it. Additionally, since elevator ascent and descent have similar signal patterns—during descent, the signal first decreases and then increases, while during ascent, it first increases and then decreases—one-second windows did not provide enough context to make a correct prediction. In Stage 8, another walking segment, the predictions were mostly correct. Only two one-second windows in the middle were misclassified: one as stairs down and another as stairs up.

Overall, the model performed well in classifying user activities in real-world scenarios, but as mentioned before, distinguishing elevator-related movements is a challenge. Nonetheless, there are promising opportunities for improvement—potentially achievable through simple additional processing, such as analyzing whether the signal shows a rise or fall prior to the current window and using that information to better determine elevator movement. This remains an area worth exploring in future studies.

## 6. Summary

In the paper, we presented a developed LSTM-based model to classify velocity sequences derived from the z-axis of linear accelerometer data captured from an Android phone sensor. The model was designed to leverage smartphone sensor data for real-time activity classification and navigation assistance, particularly in indoor environments, by distinguishing between six distinct activities: walking (10 steps), using an elevator (both up and down), climbing stairs, descending stairs, and standing.

It is important to note that the proposed method currently operates effectively only in scenarios involving the reconstruction of complete previously recorded paths. At this stage, it cannot be reliably applied when only partial segments of a route have been recorded by other users. However, addressing this limitation presents a promising direction for future research.

From an application perspective, this approach demonstrates significant potential for indoor navigation systems. Despite the relatively small dataset (480 samples) collected in various indoor environments, the model achieved

An accuracy of 75% in window-based classification using a one-second window of sensor data.An accuracy of 98.6% in full-sequence classification, where predictions from overlapping windows were aggregated using majority voting.

These results highlight the feasibility of using linear acceleration sensors for real-time activity classification and navigation assistance. This opens the door to future applications in real-world scenarios, such as guiding couriers through complex indoor spaces.

## Figures and Tables

**Figure 1 sensors-25-04673-f001:**
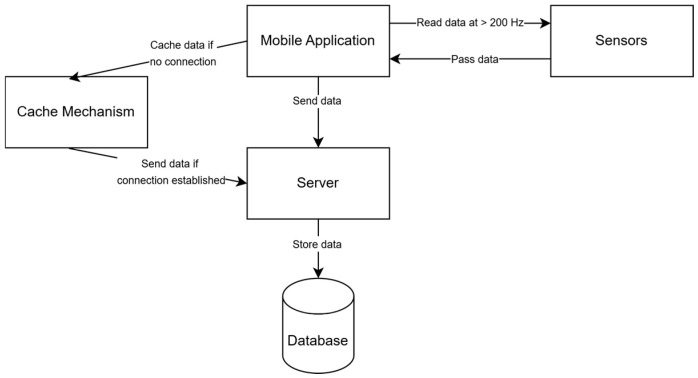
Testbed block diagram.

**Figure 2 sensors-25-04673-f002:**
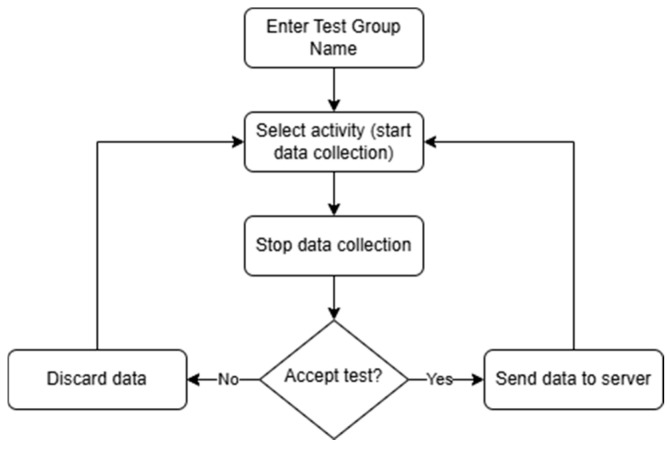
Application workflow diagram.

**Figure 3 sensors-25-04673-f003:**
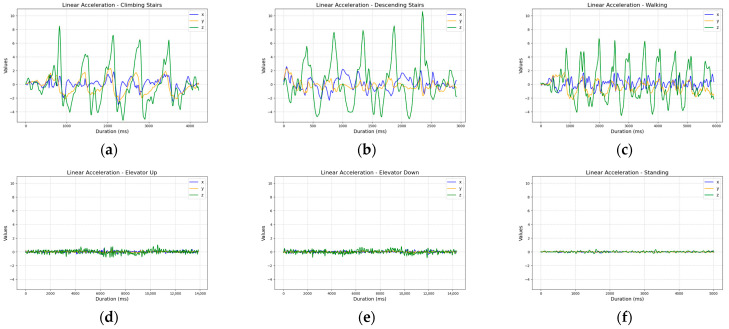
Comparison of linear acceleration sensor readings across activities: (**a**) ascending five stairs; (**b**) descending five stairs; (**c**) walking ten steps; (**d**) ascending one floor in an elevator; (**e**) descending one floor in an elevator; and (**f**) standing.

**Figure 4 sensors-25-04673-f004:**
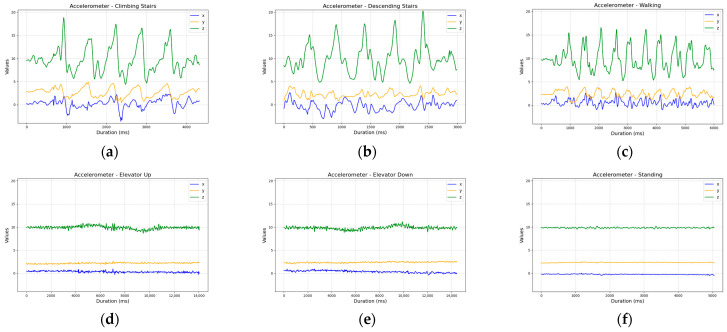
Comparison of accelerometer sensor readings across activities: (**a**) ascending five stairs; (**b**) descending five stairs; (**c**) walking ten steps; (**d**) ascending one floor in an elevator; (**e**) descending one floor in an elevator; and (**f**) standing.

**Figure 5 sensors-25-04673-f005:**
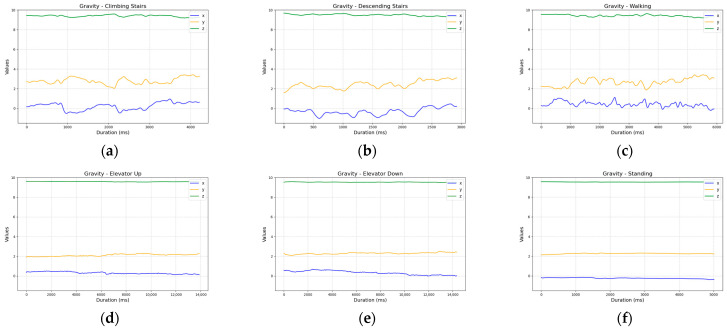
Comparison of gravity sensor readings across activities: (**a**) ascending five stairs; (**b**) descending five stairs; (**c**) walking ten steps; (**d**) ascending one floor in an elevator; (**e**) descending one floor in an elevator; and (**f**) standing.

**Figure 6 sensors-25-04673-f006:**
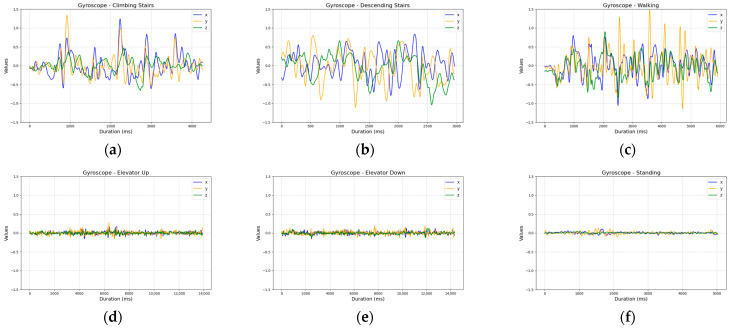
Comparison of gyroscope sensor readings across activities: (**a**) ascending five stairs; (**b**) descending five stairs; (**c**) walking ten steps; (**d**) ascending one floor in an elevator; (**e**) descending one floor in an elevator; and (**f**) standing.

**Figure 7 sensors-25-04673-f007:**
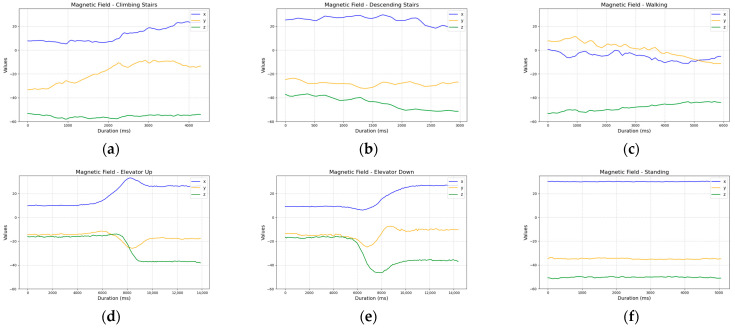
Comparison of magnetic field sensor readings across activities: (**a**) ascending five stairs; (**b**) descending five stairs; (**c**) walking ten steps; (**d**) ascending one floor in an elevator; (**e**) descending one floor in an elevator; and (**f**) standing.

**Figure 8 sensors-25-04673-f008:**
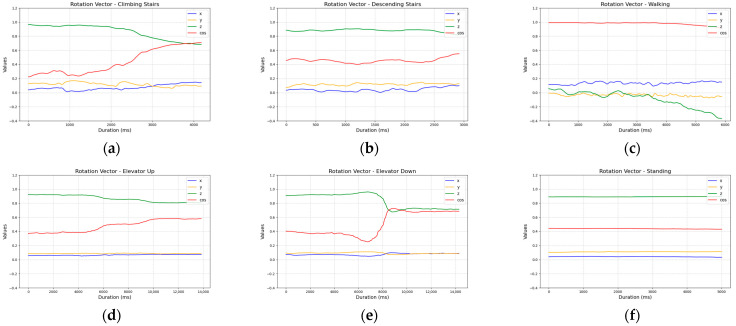
Comparison of rotation vector sensor readings across activities: (**a**) ascending five stairs; (**b**) descending five stairs; (**c**) walking ten steps; (**d**) ascending one floor in an elevator; (**e**) descending one floor in an elevator; and (**f**) standing.

**Figure 9 sensors-25-04673-f009:**
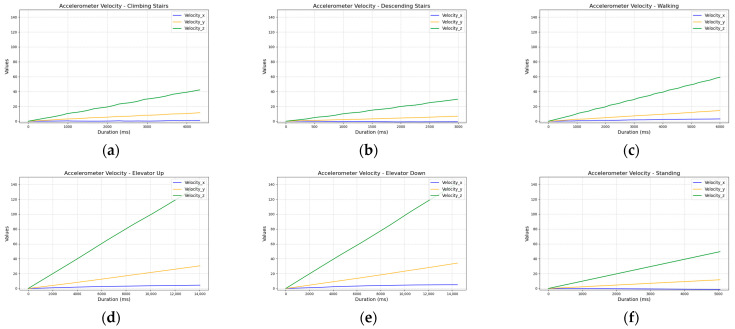
Comparison of velocity values calculated from accelerometer sensor readings across activities: (**a**) ascending five stairs; (**b**) descending five stairs; (**c**) walking ten steps; (**d**) ascending one floor in an elevator; (**e**) descending one floor in an elevator; and (**f**) standing.

**Figure 10 sensors-25-04673-f010:**
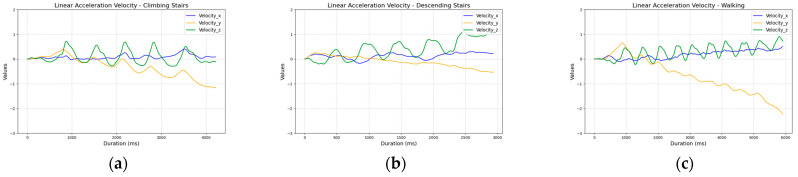
Comparison of velocity values calculated from linear acceleration sensor readings across activities: (**a**) ascending five stairs; (**b**) descending five stairs; (**c**) walking ten steps; (**d**) ascending one floor in an elevator; (**e**) descending one floor in an elevator; and (**f**) standing.

**Figure 11 sensors-25-04673-f011:**
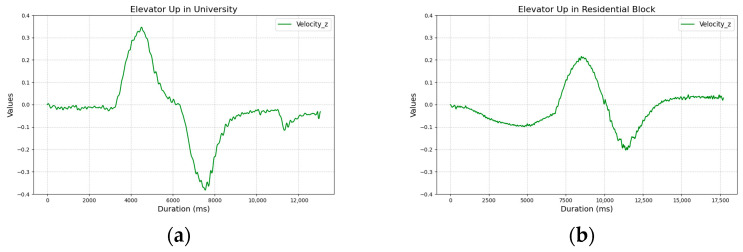
Linear accelerometer z-axis velocity data recorded in (**a**) a university elevator and (**b**) a residential block elevator.

**Figure 12 sensors-25-04673-f012:**
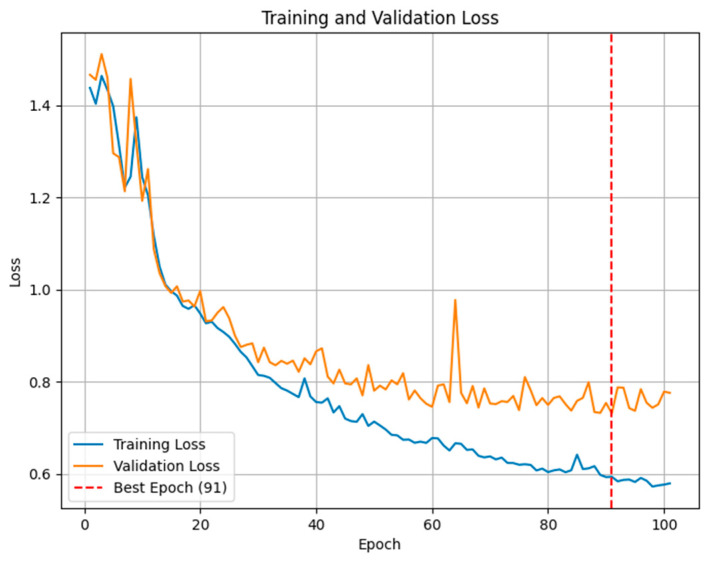
Training and validation loss over epochs, highlighting the best epoch.

**Figure 13 sensors-25-04673-f013:**
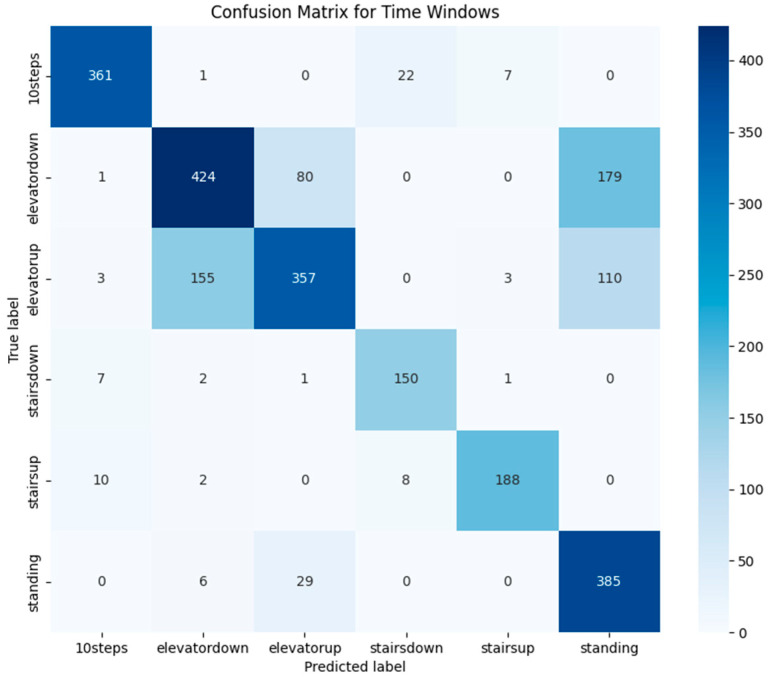
Confusion matrix for window-based classification.

**Figure 14 sensors-25-04673-f014:**
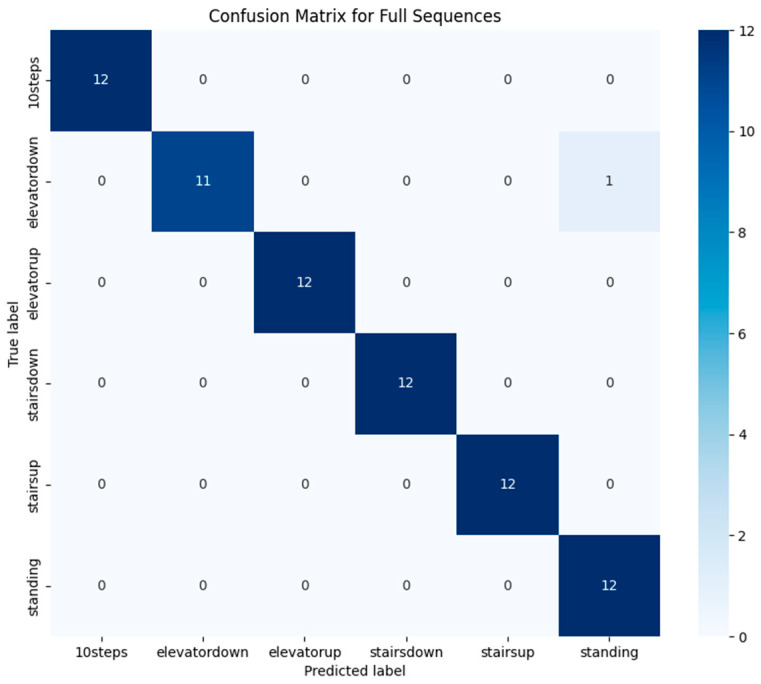
Confusion matrix for full-sequence classification.

**Figure 15 sensors-25-04673-f015:**
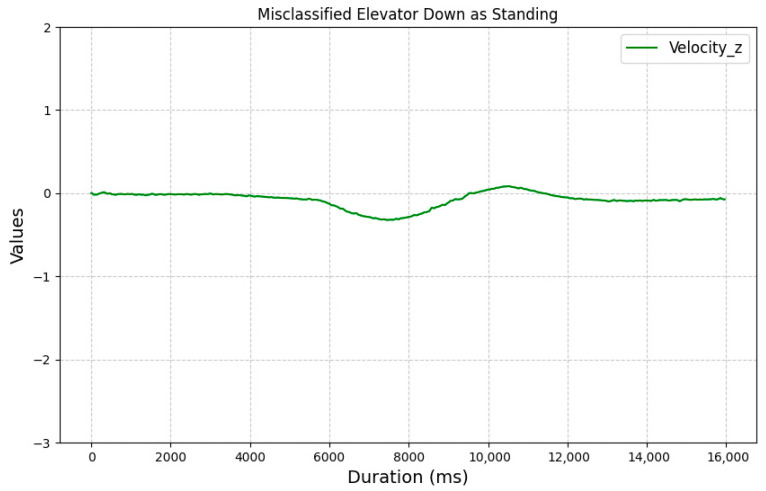
Z-axis velocity for a misclassified elevator down sequence.

**Figure 16 sensors-25-04673-f016:**
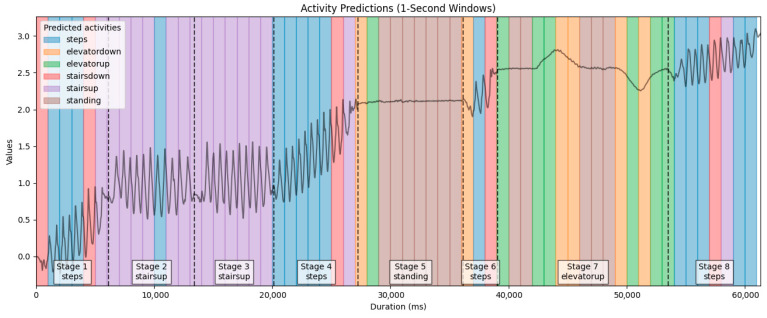
Sequence of predicted activity labels for each second in the real-world navigation test, based on velocity data from a smartphone linear acceleration.

## Data Availability

The data are available in the GitHub repository at https://github.com/borsuczek/SensorActivityRecognition (accessed on 20 June 2025).

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
