# Peer review of "Classification of User Behavior Patterns for Indoor Navigation Problem"

_sensors, 2025, doi:10.3390/s25154673_

Round 1
Reviewer 1 Report
Comments and Suggestions for Authors
1. Please describe in detail how your work differs from previous studies in the Introduction section. Also, conduct a broader survey of related work and expand the literature review accordingly.
2. The overall resolution of the figure needs to be improved.
3. In the experiment, you apply interpolation at 2 ms intervals. Isn’t this excessively fine? Please justify this choice.
4. More detailed information about the experiment is needed, such as the number of participants, the location of the experiment, and the devices used.
5. When the stair-up activity is performed in a building, how do you determine on which floor the activity occurred?
6. What are the results when using models other than LSTM? Please include a comparison.
7. Please provide a detailed description of the LSTM model, including its input, output, and architecture.
Author Response
Dear Reviewer,
On behalf of Authors of the manuscript, I would like to sincerely thank for the thorough evaluation and constructive feedback, which has significantly contributed to the improvement of our manuscript.
Below, we provide a detailed list of responses, comments, and references to the changes made in the revised manuscript, which is also attached to this correspondence.
- Please describe in detail how your work differs from previous studies in the Introduction section. Also, conduct a broader survey of related work and expand the literature review accordingly.
Thank you for this valuable remark. We appreciate your input and have made the necessary modifications in the revised version of the manuscript, as indicated in the referenced lines (26–36, 74-84). These adjustments address the concern raised and aim to improve the clarity and quality of the manuscript.
- The overall resolution of the figure needs to be improved.
Thank you for this remark. During the conversion of the Word file containing the manuscript to PDF, it is likely that the resolution of the figures was degraded. Therefore, we are including a ZIP folder with the original high-resolution generated images for the revised manuscript.
- In the experiment, you apply interpolation at 2 ms intervals. Isn’t this excessively fine? Please justify this choice.
This is a very interesting point, and indeed, during our research, we considered how best to address this issue. In the initial version of the manuscript, we did not include descriptions of our considerations in this area. Nevertheless, discussion regarding this aspect should be reflected in the revised text.
In our case, the main challenge was that Android documentation and, generally, phone APIs do not provide information on how individual sensors are sampled for a given phone model. For this reason, we decided to use the highest possible sampling frequency available from the Android API (as the Reviewer rightly noticed, this may be higher than necessary) to account for potential differences between various phone models.
To clarify this matter, we have also added additional discussion in the updated version of the manuscript in lines 258–259 and 343–345.
- More detailed information about the experiment is needed, such as the number of participants, the location of the experiment, and the devices used.
Thank you for this comment. We acknowledge that, in some places, the experimental description was too brief. We have addressed this by adding further details regarding the experimental procedure in lines 414-416 and 549-551 of the revised manuscript.
- When the stair-up activity is performed in a building, how do you determine on which floor the activity occurred?
This is indeed an intriguing comment and certainly an important point for future research. At this stage, the algorithm does not identify the specific floor on which the user is located; however, as demonstrated in the results, we are able to determine the duration of a given activity. Based on this duration, one can estimate the floor on which the user might be. It is also worth mentioning that within the proposed methodology, the focus is less on identifying the precise floor location of the user and more on determining the progress and stage of the route by matching it to a pattern. As shown in the publication, this approach enables tracking the user's movement and estimating their position.
- What are the results when using models other than LSTM? Please include a comparison.
This is a very relevant observation. I would like to add that, as part of the preliminary research for this publication, we tested various approaches for step detection and behavior classification, including those based on random forest and decision tree techniques. Ultimately, the LSTM network produced the best results in the initial stages of our research, and, as supported by a review of the literature and available technologies, its applications were the most thoroughly and comprehensively described
In this context, it should also be stated that selecting the most appropriate technology or approach in a given case depends on a range of features and the specific nature of the problem under investigation. In our opinion, this kind of issue may constitute a separate research problem altogether, and therefore a thorough analysis of this topic, in this particular study, is not feasible within the existing manuscript. Nevertheless, we agree with the comment concerning the generality of the issue of selecting research approaches, and for this reason, we have supplemented the manuscript with an additional description of these aspects in section 2.5 of the revised version.
- Please provide a detailed description of the LSTM model, including its input, output, and architecture.
Thank you for this comment. We have expanded the relevant section in Chapter 4.3, lines 458–477 of the revised manuscript to address this point.
Yours sincerely,
(on behalf of Authors) Andrzej Chybicki, PhD, Eng.
Reviewer 2 Report
Comments and Suggestions for Authors
This paper presents a developed LSTM-based model to classify velocity sequences derived from the Z-axis of linear accelerometer data captured from an Android phone sensor. However, there are a few suggestions for improvement:
- The main work of this paper is to identify the user's walking pattern based on Android sensors, so the second part should focus on the research progress of existing recognition methods, rather than indoor positioning methods.
- Pixels of all figures is too low to read. The header and footer contain a lot of white space and need to be reformatted.
- Were the data from Figure 3 to Figure 10 collected at the same time? How long is the collection time? Does each sensor collect at the same frequency? What is the frequency?
- Each variable needs to be explained when it first appears in all equation.
- “These observations confirmed that processed data from the z-axis of the linear acceleration sensor could serve as a solid foundation for further activity analysis and classification.” How did the author come to this conclusion? The authors should give a clearer explanation and evidence.
- Figure 17 lacks a statistical table on recognition accuracy. The experimental sequence time is only 1 minute, which is too short to verify the recognition accuracy of the proposed model, and more sequence time is required, such as 15 minutes.
Author Response
On behalf of Authors of the manuscript, I would like to sincerely thank for the thorough evaluation and constructive feedback, which has significantly contributed to the improvement of our manuscript.
Below, we provide a detailed list of responses, comments, and references to the changes made in the revised manuscript, which is also attached to this correspondence.
1. The main work of this paper is to identify the user's walking pattern based on Android sensors, so the second part should focus on the research progress of existing recognition methods, rather than indoor positioning methods.
Thank you to the Reviewer for this comment. Indeed, the state of the art regarding activity recognition methods was not sufficiently emphasized in the first version of the manuscript. Therefore, we have decided to add an additional subsection 2.5 entitled “Pattern-based approaches,” which addresses the suggestion mentioned (lines 154–199 in the revised version of the manuscript).
2. Pixels of all figures is too low to read. The header and footer contain a lot of white space and need to be reformatted.
Thank you for this remark. During the conversion of the Word file containing the manuscript, it is likely that the resolution of the figures was degraded. Therefore, we are including a ZIP folder with the original high-resolution generated images for the revised manuscript.
3. Were the data from Figure 3 to Figure 10 collected at the same time? How long is the collection time? Does each sensor collect at the same frequency? What is the frequency?
We thank the Reviewer for this remark. In the examples presented in Figures 3 and 10, the time ranges for the individual tests correspond to each other; that is, the time range shown in Fig. 3a for the ‘Climbing Stairs’ test matches the range in Fig. 10a, and similarly for Figs 3b and 10b. The test duration is presented in milliseconds. The issue of sampling frequency results from the specifics of the Android API operation for different versions and models of phones. Since it was not described precisely earlier, we have included an additional explanation in lines 258–259 and 343–345 of the revised version of the manuscript.
4. Each variable needs to be explained when it first appears in all equation.
Thank you for this observation. In the course of revising the manuscript, we have introduced descriptions of variables that were previously omitted, including those in equations (8) in lines 394-395 and Eq.(9) (lines 445-448) of the revised version of the manuscript.
5. “These observations confirmed that processed data from the z-axis of the linear acceleration sensor could serve as a solid foundation for further activity analysis and classification.” How did the author come to this conclusion? The authors should give a clearer explanation and evidence.
Thank you for this remark. We have included additional discussion on this topic in lines 425–430 of the revised version of the manuscript.
6. Figure 17 lacks a statistical table on recognition accuracy. The experimental sequence time is only 1 minute, which is too short to verify the recognition accuracy of the proposed model, and more sequence time is required, such as 15 minutes.
We thank the Reviewer for this observation. Indeed, in this section of the article, we did not perform a statistical error analysis of the algorithm's operation for the described case, as in our opinion, it would to some extent replicate the actions and methodology described in Section 5.1 of the manuscript. Although our manuscript has a theoretical and experimental nature, the purpose of the actions described in Section 5.2, to which this comment refers, was to demonstrate to what extent our approach could practically work and to present it in an accessible manner. Of course, we agree with the Reviewer that such an analysis would be a valuable element of future studies, especially those focused on practical applications and evaluation of the described methods. We thank the Reviewer for this suggestion and will certainly take it into account in our future research on the described topic.
Regarding the second part of the question, we would like to emphasize that in our research, the goal was to focus on short paths that are most frequently taken, for example, by couriers or food delivery providers, and these use cases were the scenarios we examined to assess the usefulness of our model. As the Reviewer rightly pointed out, our model is designed with indoor navigation in mind, which is intended to last no more than 2–3 (maximum 4) minutes. For this reason, the results presented in Section 5.2 were intended solely as an initial practical verification of the model proposed by us for the aforementioned use cases. Nevertheless, we believe the comment is valid, and as a result, we have decided to add an additional discussion in lines 562–571 of the revised version of the manuscript.
On case of any questions, I remain at your disposal
(On behal of Authors) Andrzej Chybicki, PhD. Eng.
Reviewer 3 Report
Comments and Suggestions for Authors
User behavior patterns can definitely be used for indoor navigation, which is beyond doubt and will play a significant role, especially in the absence of absolute positioning means such as wireless signals. In this paper, a series of routine activity recognition experiments were conducted using self-developed application software, with high accuracy, which can support the results and conclusions. However, there are still some problems that need to be solved before considering publication.
- The assumption is that someone has already walked a special route inside the building once, and the others can conduct indoor navigation with the help of the previous route and behavior classification. But if that person only walked a part of the building and did not cover the route of the pedestrians behind, would the method also work in this case? Please give explanations for the discussion.
- At the beginning of the Section 2, several indoor positioning techniques are introduced without any references, such as WiFi BLE magnetic field. 'WhereArtThou','IDWPSOInLoc', 'BLE-based indoor localization', 'BleHe', 'MSPos','Ped-Mag-ODO'. are suggested.
- In the section 2.1, its accuracy is affected by many factors. Are there references to support the conclusion?
- It is necessary to clarify in the manuscript how the sensor data and duration corresponding to different activities or behaviors are extracted?
- Another several machine learning methods are suggested to evaluate the effectiveness of the proposed method using the same dataset.
Author Response
Dear Reviewer,
On behalf of Authors of the manuscript, I would like to sincerely thank for the thorough evaluation and constructive feedback, which has significantly contributed to the improvement of our manuscript.
Below, we provide a detailed list of responses, comments, and references to the changes made in the revised manuscript, which is also attached to this correspondence.
1. The assumption is that someone has already walked a special route inside the building once, and the others can conduct indoor navigation with the help of the previous route and behavior classification. But if that person only walked a part of the building and did not cover the route of the pedestrians behind, would the method also work in this case? Please give explanations for the discussion.
Reviewer 3 has made a valid observation that was not addressed in the manuscript. The proposed method is designed solely for route reconstruction, which assumes that the specific path must have been fully traversed by a previous user. In cases where only a part of the route has been recorded, this method will not be applicable for indoor navigation purposes. This is indeed a notable point and an interesting avenue for future research; however, at this stage of study, such scenarios are not considered.
To accommodate this valid observation from the reviewer, we have added the corresponding discussion in lines 619-623 of the manuscript.
2. At the beginning of the Section 2, several indoor positioning techniques are introduced without any references, such as WiFi BLE magnetic field. 'WhereArtThou','IDWPSOInLoc', 'BLE-based indoor localization', 'BleHe', 'MSPos','Ped-Mag-ODO'. are suggested.
We thank the Reviewer for this insightful remark. In accordance with the suggestion, we have supplemented the set of utilized references with items [14–18] in the revised version of the manuscript.
3. In the section 2.1, its accuracy is affected by many factors. Are there references to support the conclusion?
We thank the Reviewer for this observation—indeed, this issue requires clarification. In response to the Reviewer’s suggestion, we have added two bibliographic entries ([22] and [23]) addressing this matter.
4. It is necessary to clarify in the manuscript how the sensor data and duration corresponding to different activities or behaviors are extracted?
We thank the Reviewer for this valuable comment. The duration of individual activities was marked during the data collection process. In practice, for each type of activity during the collection of test data, we specified what kind of activity the test concerned. This issue was addressed in Section 3.3,however, after thorough analysis of Reviewer’s suggestion, we decided to clarify this description further by introducing modifications in lines 264-265 of the manuscript.
5. Another several machine learning methods are suggested to evaluate the effectiveness of the proposed method using the same dataset.
This is a very pertinent comment from the Reviewer, and indeed, within the scope of our literature analysis and research activities, there exist numerous machine learning methods that could potentially address this issue. In our preliminary research, we also explored approaches based on the RandomForest algorithm as well as other available simpler neural network architectures. However, we ultimately decided not to include the results of these initial studies, as they produced significantly poorer results than those presented in our manuscript.
Nevertheless, we fully concur with the Reviewer’s suggestion, and therefore we have expanded the discussion on this topic, including relevant elaborations in lines 177–185, and we have added references to the bibliography, among others in items [7], [12], [15], [18], and [19].
In case of any questions, I remain at youd disposal.
(on behalf of Authors) Andrzej Chybicki, PhD, Eng.
Round 2
Reviewer 1 Report
Comments and Suggestions for Authors
Thank you for your revision. I have no more comments.
Author Response
Dear Reviewer,
Thank you for taking the time to review our manuscript and for your comments, which have contributed to the revision process. We appreciate your input and attention to detail.
Yours sincerely,
(on behalf of The Authors) Andrzej Chybicki, PhD Eng.
Reviewer 2 Report
Comments and Suggestions for Authors
I have downloaded the file, but before beginning my scientific assessment, I noticed that the current version is not suitable for review due to critical formatting issues:
- The document contains unresolved comments (lines 344 and 560). A clean version is required for an objective review.
- The line number is incorrect in your responses, which prevents me from providing precise, actionable feedback.
For these reasons, I am unable to proceed with the review at this time. Could you please submit a properly formatted manuscript that is a clean version (no comments from Microsoft Word) and has correct line number in your responses?
Author Response
Dear Reviewer,
Thank you for your feedback, and we regret that the formatting issues have hindered your analysis. Enclosed with this revision, we are submitting a cleanly formatted manuscript (all tracked changes have now been accepted). Additionally, to facilitate your review, we are providing updated line numbers corresponding to the changes addressed in our responses. For clarity in the review process, below we are including the previously provided responses with minor modifications based on the suggestions received from the third Reviewer.
Comment 1. The main work of this paper is to identify the user's walking pattern based on Android sensors, so the second part should focus on the research progress of existing recognition methods, rather than indoor positioning methods.
Response 1:
Thank you to the Reviewer for this comment. Indeed, the state of the art regarding activity recognition methods was not sufficiently emphasized in the first version of the manuscript. Therefore, we have decided to add an additional subsection 2.6 entitled “Pattern-based approaches,” which addresses the suggestion mentioned (lines 157–184 in the revised version of the manuscript).
Comment 2. Pixels of all figures is too low to read. The header and footer contain a lot of white space and need to be reformatted.
Respone 2:
Thank you for this remark. During the conversion of the Word file containing the manuscript, it is likely that the resolution of the figures was degraded. Therefore, we have included a ZIP folder with the original high-resolution generated images for the revised manuscript.
Comment 3. Were the data from Figure 3 to Figure 10 collected at the same time? How long is the collection time? Does each sensor collect at the same frequency? What is the frequency?
Response 3:
We thank the Reviewer for this remark. In the examples presented in Figures 3 and 10, the time ranges for the individual tests correspond to each other; that is, the time range shown in Fig. 3a for the ‘Climbing Stairs’ test matches the range in Fig. 10a, and similarly for Figs 3b and 10b. The test duration is presented in milliseconds. The issue of sampling frequency arises from the specific operation of the Android API on different versions and models of smartphones, which was taken into account in our research. In some cases the phone documentation doesn’t provide the information on maximum sampling frequency, however each dataset contains the frequency used to register the data. Since this issue was not described precisely earlier, we have included an additional explanation in lines 243-244 and 328-331 of the revised version of the manuscript.
Comment 4. Each variable needs to be explained when it first appears in all equation.
Response 4:
Thank you for this observation. In the course of revising the manuscript, we have introduced descriptions of variables that were previously omitted, including those in equations (8) in lines 380-381 and Eq.(9) (lines 431-433) of the revised version of the manuscript.
Comment 5. “These observations confirmed that processed data from the z-axis of the linear acceleration sensor could serve as a solid foundation for further activity analysis and classification.” How did the author come to this conclusion? The authors should give a clearer explanation and evidence.
Response 5:
Thank you for this remark. We have included additional discussion on this topic in lines 410-415 of the revised version of the manuscript.
Comment 6. Figure 17 lacks a statistical table on recognition accuracy. The experimental sequence time is only 1 minute, which is too short to verify the recognition accuracy of the proposed model, and more sequence time is required, such as 15 minutes.
Response 6:
We thank the Reviewer for this observation. Indeed, in this section of the article, we did not perform a statistical error analysis of the algorithm's operation for the described case, as in our opinion, it would to some extent replicate the actions and methodology described in Section 5.1 of the manuscript. Although our manuscript has a theoretical and experimental nature, the purpose of the actions described in Section 5.2, to which this comment refers, was to demonstrate to what extent our approach could practically work and to present it in an accessible manner. Of course, we agree with the Reviewer that such an analysis would be a valuable element of future studies, especially those focused on practical applications and evaluation of the described methods. We thank the Reviewer for this suggestion and will certainly take it into account in our future research on the described topic.
Regarding the second part of the question, we would like to emphasize that in our research, the goal was to focus on short paths that are most frequently taken, for example, by couriers or food delivery providers, and these use cases were the scenarios we examined to assess the usefulness of our model. As the Reviewer rightly pointed out, our model is designed with indoor navigation in mind, which is intended to last no more than 2–3 (maximum 4) minutes. For this reason, the results presented in Section 5.2 were intended solely as an initial practical verification of the model proposed by us for the aforementioned use cases. Nevertheless, we believe the comment is valid, and as a result, we have decided to add an additional discussion in lines 540-549 of the revised version of the manuscript.
Reviewer 3 Report
Comments and Suggestions for Authors
All my concerns have not been answered and modified well. The explanations are not detailed enough and the revisions are relatively few. None of them is satisfactory. Especially, the 1st issue is extremely important and is the key to supporting the innovation of the manuscript. I feel so sorry that the author's reply is No. I'm skeptical about the innovativeness of the method.
Author Response
Dear Reviewer,
I am truly sorry to hear that our responses have not met your expectations—we have made every effort to address your suggestions and comments, most of which we agree with. In response to your suggestions and those of other reviewers, we are providing the following answers, modifications, and changes in the new revision of the manuscript, with the hope that they are now sufficiently clear.
I. (Round 1) Comment 1:
The assumption is that someone has already walked a special route inside the building once, and the others can conduct indoor navigation with the help of the previous route and behavior classification. But if that person only walked a part of the building and did not cover the route of the pedestrians behind, would the method also work in this case? Please give explanations for the discussion.
(Round 2) Comment 1:
Especially, the 1st issue is extremely important and is the key to supporting the innovation of the manuscript. I feel so sorry that the author's reply is No. I'm skeptical about the innovativeness of the method.
Updated response 1:
In the introduction to the manuscript, the motivation section of the new revision, as well as in our previous response, we have provided a description of what, in our view, constitutes the innovativeness of the proposed solution. Our approach is primarily intended for use by courier companies delivering food and parcels to locations such as apartment complexes and multi-store residential buildings. We believe we are the first to leverage the repeatability of patterns identified from multiple couriers using the same route, a feature not seen in current solutions. At this stage of the research, we have not yet focused in detail on how to implement this solution in practice or its potential additional functionalities and extensions. One such extension could be precisely the scenario proposed by the Reviewer, i.e., reconstructing the route in cases where the courier has not reached the final destination. However, we believe that such situations will occur relatively rarely in practice— in most cases, couriers delivering a parcel walk the entire route from the beginning (i.e., the entrance to the building) to the end (the designated apartment).
In our assessment, this approach is innovative also because, for the first time in the world, it potentially enables courier companies to use data from previous deliveries to improve deliveries carried out by new employees and to expedite the delivery process.
We also do not view our response to your suggestions as a dismissal—instead, we see your suggestion as a valuable avenue for future research, though it extends beyond the present scope of our manuscript.
As earlier mentions (round 1) to accommodate this valid observation, we have added the corresponding discussion in lines 590-594 of the revised version 2 of the manuscript, where we emphasize the limitation of our approach at current stage.
II. Responses to suggestions 2-5
Below we also update responses form previous round with updated line numbers in the revised version of the manuscript. The current version of the manuscript is free of track changes.
Comment 2: At the beginning of the Section 2, several indoor positioning techniques are introduced without any references, such as WiFi BLE magnetic field. 'WhereArtThou','IDWPSOInLoc', 'BLE-based indoor localization', 'BleHe', 'MSPos','Ped-Mag-ODO'. are suggested.
Response 2:
We thank the Reviewer for this insightful remark. In accordance with the suggestion, we have supplemented the set of utilized references with items [14–22] in the revised version of the manuscript, including articles containing the suggested keywords (‘WhereArtThou’, ‘IDWPSOInLoc’, ‘BLE-based indoor localization’, ‘BleHe’, ‘MSPos’, ‘Ped-Mag-ODO’). Additionally, we have added Section 2.3: Wi-Fi Navigation (lines 114-128), where we describe technologies presented in the articles ‘WhereArtThou’ and ‘IDWPSOInLoc’. Previously, we had sections covering inertial, BLE, magnetic field, and dead reckoning methods, but we did not include Wi-Fi-based approaches.
Comment 3: In the section 2.1, its accuracy is affected by many factors. Are there references to support the conclusion?
Response3:
We thank the Reviewer for this observation—indeed, this issue requires clarification. In response to the Reviewer’s suggestion, we have added two bibliographic entries ([26] and [27]) addressing this matter.
Comment 4: It is necessary to clarify in the manuscript how the sensor data and duration corresponding to different activities or behaviors are extracted?
Response 4:
We thank the Reviewer for this valuable comment. The duration of individual activities was marked during the data collection process. In practice, for each type of activity during the collection of test data, we specified what kind of activity the test concerned. This issue was addressed in Section 3.3,however, after thorough analysis of Reviewer’s suggestion, we decided to clarify this description further by introducing modifications in lines 237 and 246-250 of the manuscript.
Comment 5: Another several machine learning methods are suggested to evaluate the effectiveness of the proposed method using the same dataset.
Response 5:
This is a very pertinent comment from the Reviewer, and indeed, within the scope of our literature analysis and research activities, there exist numerous machine learning methods that could potentially address this issue. In our preliminary research, we also explored approaches based on the RandomForest algorithm as well as other available simpler neural network architectures. However, we ultimately decided not to include the results of these initial studies, as they produced significantly poorer results than those presented in our manuscript.
Nevertheless, we fully concur with the Reviewer’s suggestion, and therefore we have expanded the discussion on this topic, including relevant elaborations in lines 163-174, and we have added references to the bibliography.
Your sincerely,
(on behal of Authors) Andrzej Chybicki, PhD. Eng.
Round 3
Reviewer 2 Report
Comments and Suggestions for Authors
In this new version, the authors improve the article considering the suggestions given by revisors. I do not have any additional comments.
Reviewer 3 Report
Comments and Suggestions for Authors
Both the response and modification are detailed. The manuscript has been significantly improved.